# Communication of the Diagnosis of Spinal Muscular Atrophy in the Views of Patients and Family Members, a Qualitative Analysis

**DOI:** 10.3390/ijerph192416935

**Published:** 2022-12-16

**Authors:** Isabella Araujo Mota Fernandes, Renata Oliveira Almeida Menezes, Guilhermina Rego

**Affiliations:** 1Faculty of Medicine, University of Porto, 4200-319 Porto, Portugal; 2Law Department, Federal University of Rio Grande do Norte, Caico 59300-000, Brazil

**Keywords:** spinal muscular atrophy, physician-patient relations, communication, diagnosis

## Abstract

Introduction: Communicating the diagnosis of a genetic and neurodegenerative disease, such as spinal muscular atrophy (SMA), requires a transmission centered on the patient and/or the family caregiver, ensuring autonomy to those involved and strengthening the doctor–patient relationship. Objective: Analyzing the communication of the SMA diagnosis from the perspective of patients and family members. Methods: This qualitative study was developed through semi-structured interviews, via teleconsultation. The analysis was developed by systematically condensing the answers and synthesizing them into four thematic axes (clarification of the diagnosis, communication of the prognosis, affective memory related to the event, and advice to physicians). Results and discussion: Twenty-nine patients with SMA and 28 family caregivers of people with this condition, from all regions of Brazil, reported that individualized, clear, honest, and welcoming communication, emphasizing positive aspects, in the presence of family members and with the possibility of continuous monitoring, was important to meeting their communication needs. A lack of empathy, monitoring and guidance, and estimating life expectancy resulted in negative experiences. Conclusions: The communication needs of patients and family members described during the clarification of the diagnosis and prognosis of SMA predominantly involve empathic factors related to the attitude of the attending physician throughout the evolution of the disease. Future research evaluating other neurodegenerative diseases and the development of research protocols are important to improving communication between physicians, patients, and family members.

## 1. Introduction

The importance of communication between doctors and patients, for the benefit of health care, has been described since the beginning of medicine [1]. In this sense, respect for people, beneficence, and justice were defined as ethical principles that should guide biomedical conduct during the Belmont Report, in 1978. Beauchamp and Childress followed up in 1979, who established beneficence, non-maleficence, autonomy, and justice as general principles of medical practice [2,3].

Traditional paternalistic medicine, in which the opinion of the physician prevails, is still common in clinical practice [4,5], reducing the freedom of opinion and autonomy of patients [2,3]. Respect for autonomy is related to the veracity and clarification of the information provided, and the request for consent from agents capable of conscious and voluntary decision-making, be they patients or their legal guardians [2,6,7]. By understanding the impacts of their decisions, autonomous beings have the freedom to decide according to their beliefs, perspectives, and values, even against the opinions of the attending physicians [4,6].

Spinal muscular atrophy (SMA) is a rare, genetic, neurodegenerative, and disabling disease, whose symptoms can appear from early childhood to early adulthood. It results in the dysfunction and death of lower motor neurons, leading to decreases in appendicular, axial, and medullary strength, culminating in difficulty in swallowing, speaking, breathing, and maintaining cognitive integrity and sensory and sensorial functions [8].

Respecting autonomy and adequate communication of the diagnosis and prognosis of neurodegenerative and disabling diseases, such as SMA, impact the relationships among the physicians, patients, and family caregivers; the mental health of those involved; and the therapeutic follow-up [8,9,10]. Accomplishing said things helps to avoid anguish and suffering [9,11], which can short and long-term repercussions for patients and family caregivers with genetic disorders [12,13].

In recent years, dissatisfaction has been observed regarding emotional support, time spent in medical consultations, and the nature and quality of information on neurodegenerative diseases [10]. The need for communication standards centered on patients [5] and/or their families [9,12] addresses respect and protection of autonomy [6].

This research aims to understand which aspects of communicating the SMA diagnosis are considered positive and negative experiences in the view of patients and their family caregivers, to improve the healthcare experience, since there is a gap in the literature related to communicating this diagnosis, causing moderate to intense stress among neurologists [10].

## 2. Methodology

This is a cross-sectional study performed between September 2020 and March 2022, approved by the ethics and research committee of the Lauro Wanderley University Hospital, Federal University of Paraiba, under opinion number 5,176,679. Patients with SMA or their family members, living in Brazil, over 18 years of age, and without a history of cognitive impairment, were invited to participate. Sampling was of the simple random type; and contacts were obtained during the study period, through the Regional and National Association of Carriers of Rare Diseases, Association of Neuromuscular Diseases of Paraiba (DOENMUS), Friends of SMA (AAME), and Association of Families and Friends of Carriers of Neuromuscular Diseases (DONEM).

Responses were obtained from patients and family members from public and private services in equal proportions. The reports described range from the pre-genetic testing era to the present day. Currently, genetic tests are available in the private system and some specialized public services. During this research, drugs to treat SMA were acquired only through legal means, but there are three of them (nusinersen, onasemnogene abeparvovec-xioi, and risdiplam) in the process of incorporation.

After agreeing with the terms of free, informed, and clarified consent, patients and family caregivers received a call (teleconsultation) with an average duration of 15 min, to respond to a semi-structured interview developed by a neurologist with expertise in neuromuscular diseases. The other authors, a specialist in medical law and bioethics, and a researcher specialized in palliative care and bioethics, participated in the analysis and interpretation of the data, to reduce the bias related to the examiner, thereby maintaining reliability, triangulation, and credibility [14]. Suggestions and criticism were sent by email to the first author, who conducted the final review and evaluation.

Patients and family caregivers answered the following questions: “How did you receive the SMA diagnosis?”, “What was negative in communicating the diagnosis?”, “What was positive in communicating the diagnosis?”, “How would you like to have received the SMA diagnosis?”, “What advice would you send to health professionals who participated in your journey through the diagnosis of SMA?”, and finally, “If there is anything about the moment of receiving the SMA diagnosis that you want to tell me, I am eager to hear about it”.

A qualitative analysis was performed through the systematic condensation of the answers. All responses were recorded, transcribed literally (in 106 pages), decontextualized, divided into units of meaning, analyzed, interpreted with the re-contextualization of sentences, condensation of repeated answers, and then organized into four thematic axes [14,15] that reveal how the interviewees were informed about the SMA diagnosis—namely: clarification of the diagnosis, communication of the prognosis, affective memory related to the process of diagnosis, and advice to physicians who communicated the diagnosis and prognosis. The survey followed the criteria suggested by the Standards for Reporting Qualitative Research [15].

## 3. Results

Fifty-seven people from all regions of Brazil were interviewed. Twenty-nine were SMA patients, and twenty-eight were family caregivers. All family caregivers present at the time of diagnosis were mothers. The mean age of the patients was 32.86 years (range: 19–46) and the mean age of the mothers was 39.43 years (range: 24–53). There was a predominance of SMA type 3 (49%), followed by types 2 (39%), 1 (10%), and 4 (2%) (Table 1). Until the definitive diagnosis, 36 (63%) patients and family caregivers heard opinions from three or more doctors in public and private services. The reports of positive and negative experiences related to the thematic axes analyzed during the semi-structured interview are described in Table 2.

## 4. Discussion

There are five clinical forms of SMA, classified as types 0, 1, 2, 3, and 4, with different prognoses. Type 1 is the most frequent, but with early death, which justifies the absence of patients with this diagnosis and its low-rate prevalence in the family group. The absence of SMA type 0 is justified by its rarity, underdiagnosis, and low life expectancy. Forty-nine percent of respondents had SMA type 3 or were the mother of a patient with this phenotype, whose life expectancy was equal to that of the general population; and 39% presented type 2, whose expectation is to reach adulthood [8].

The semi-structured interview allowed participants to remember the moment they received the SMA diagnosis without interference from the examiner, reporting aspects they considered important and that impacted the communication process positively and negatively, as discussed below:

### 4.1. Clarification of the Diagnosis

Patients and mothers who received the diagnosis without explanation of its clinical meaning, using inappropriate terms or via written documents, such as email or letters, reported feelings of anguish, despair, and abandonment because they did not understand the technical language. They sought information from unreliable sources or delayed the start of treatment, in addition to not feeling embraced in the face of the negative impact of the diagnostic conclusion (see items 1.a–m, in Table 2).

During the process of diagnosing SMA, clarifying the announcement is essential and should be performed as early as possible [12,16,17,18], to facilitate the exchange of information and the sharing of decisions, thereby strengthening the reliability in the physician–patient relationship and improving the care experience [12,19,20]. However, breaking bad news is not a simple transfer of information. One must understand who will receive the news and what their experiences, expectations, and knowledge about the subject are [18,21,22], in order to meet their information needs [23,24] (see items 1.A–J, from Table 2). 

Thus, the diagnosis must be communicated by a doctor in an appropriate environment, using clear vocabulary, without ambiguities and jargon, while giving some time for patients and families to clarify doubts [12,17,18,24] to avoid the “feeling of abandonment” and dissatisfaction [10,20] from causing “psychological trauma” [8,12] or post-traumatic stress syndrome [25].

Failures in communication can generate feelings of insecurity, frustration, anger, and anxiety [26]. Clarifying doubts throughout the course of the disease, providing reading materials and continuing education, favored the understanding of limitations, the establishment of goals, and decision making, in addition to strengthening the bond with the physician [16,20,21,25]. Communicating the truth has had a positive impact on the vitality of patients [27]. Thus, announcing SMA must be associated with short, medium, and long-term guidelines, and the support of an interdisciplinary team [16,17,28].

Scheduling an early return appointment is important because, at first, the information may not be well assimilated [17,18]. Thus, guiding the recording of doubts to be discussed in a second moment [17] may reduce inadequate decision making because of vulnerability and misinformation [29]. Omitting the diagnosis and distorted, controversial, incomplete, and late information negatively influence the acceptance and coping of family members [22].

### 4.2. Communicating the Prognosis

Anticipating the prognosis and estimating life expectancy had negative impacts on communication (see items 2.a–i, in Table 2). An individualized prognosis communication guide suggests knowing the needs of individual information regarding life expectancy, and clarifying that, although there is a statistical estimate, there is also a variability of progression, with different patients presenting better or worse clinical evolutions [19].

Multiple consultations may be necessary to understand the information needs of patients and their family members [11,21], in addition to their cultural, social, and religious perceptions. Emphasizing positive aspects of the disease, along with future possibilities and planning goals, help in the moment of communicating the prognosis (see items 2.A–D, in Table 2). The participation of family members, the right not to know when requested, and the support of an interdisciplinary team have a positive impact on the communication of prognosis and discussions on life expectancy, improving decision-making and future planning [28].

### 4.3. Affective Memory Related to the Time of Diagnosis

Emotional support with an empathic posture involves not only what is said and written, but non-verbal aspects, such as looks, smiles, touch, nods, and the posture of professionals in the face of the reaction from the patient. Crying, silence, denial, fear, or anger must be understood and welcomed [18,24]. Although empathy is considered a basic communication skill [19] and is present in protocols for breaking bad news [18,24], its absence has been described during the transmission of the diagnosis of neurodegenerative diseases [10].

Empathy, or the lack of it, was remarkable throughout the communication process, from the clarification of the diagnosis and communication of the prognosis to the medical follow-up. Positive empathic experiences were expressed by terms such as attention, care, calmness, kindness, support, welcoming, sensitivity, affection, and serenity (see items 3.A–G, in Table 2). The lack of empathy proved to be an important communication barrier among the interviewees, causing negative feelings such as insensitivity and helplessness [22] (see items 3.a–l, in Table 2).

Communication barriers compromise the care experience and general quality of life, due to poor understanding of diagnosis and prognosis, therapeutic limitations, and possible incapacity or incompetence in decision making [9,20,28,30]. Despite the barrier related to staff’s insecurity in transmitting uncertainties related to the evolution of diseases, fearing a negative impact on the lives of the sick, honesty, the ability to give hope without deceiving, empathy, and individualization of the case empower the patient, positively impacting the lives of those involved [19,28,30]. Another relevant factor is the number of people in the communication environment, affecting privacy in the moment of facing bad news [12,17] (see items 3.l, in Table 2).

### 4.4. Advice to Doctors Who Reported the Diagnosis

The advice section expresses the importance of empathic and individualized communication and listening to and understanding the needs of patients and/or their mothers at each consultation; clarification of the illness; continued education through therapeutic guidelines; prognosis being given in a clear and honest way such that hope and faith are not taken away, but, at the same time, such that unrealistic expectations are not generated (see items 4.A–L, in Table 2).

Individual communication needs must be met; positive aspects should be emphasized in the presence of family members; and the possibility of continuous monitoring for clarifying future doubts should be mentioned. The lack of empathy proved to be an important communication barrier, along with the anticipation of prognosis, estimating life expectancy, and the absence of guidance and follow-up [30]. 

## 5. Conclusions

This study has shown that individual and family communication needs during the clarification of the diagnosis and prognosis of SMA predominantly involve empathic factors throughout the evolution of the disease. Negative experiences in communicating the diagnosis of SMA, described by patients and their mothers, warn of the need to improve and adapt techniques and protocols for breaking bad news for neurologists [29,30,31]. In order to strengthen the millenary relationship between physicians and the patient, we must favor the application of ethical principles in medical practice, which should consequently improve short and long-term therapeutic responses and adherence to ongoing care for these patients. This research should be expanded to other neurodegenerative diseases, and be used as a basis for developing and validating research protocols to improve communication between physicians and patients and families.

## 6. Limitations

Among the limitations, there could have been a bias regarding the memory of bad news bias contributing to the predominance of negative memories. The higher prevalence of SMA types 2 and 3 in the sample could have generated a life expectancy bias, and the predominance of females, although expected, as mothers are culturally more participatory in monitoring their children, may have caused a bias related to sensitivity in the perception of information. The absence of a validated research tool for this purpose is another limitation of this research. Future studies should adapt and validate this research tool to conduct analyses of both quantitative and qualitative types.

## Figures and Tables

**Table 1 ijerph-19-16935-t001:** Demographic profile of SMA patients and family caregivers who participated in this study.

Variables	SMA Patients		SMA Family Caregivers	
	*n*	%	*n* %	
**SEX**				
Male	12	41.4%	0	0%
Female	17	58.6%	28	100%
**AGE**				
≤30 years old	15	51.7%	3	10.7%
>30 years old	14	48.3%	25	89.3%
**COLOR/RACE**				
Yellow	0	0%	2	7.1%
White	21	72.4%	13	46.5%
Black	0	0%	2	7.1%
Brown	8	27.6%	11	39.3%
**EDUCATION**				
Postgraduate studies	4	13.9%	5	17.9%
Higher education	9	31%	8	28.6%
Incomplete high school	2	6.9%	1	3.6%
Complete high school	12	41.4%	9	32.1%
Incomplete primary education	1	3.4%	2	7.1%
Complete primary education	1	3.4%	3	10.7%
**REGION FROM BRAZIL**				
Center-West	1	3.4%	1	3.6%
Northeast	12	41.4%	18	64.3%
North	1	3.4%	0	0%
Southeast	11	38.0%	5	17.9%
South	4	13.8%	4	14.2%
**Family income**				
<3 minimum wages *	16	55.2%	18	64.3%
between 3–5 minimum wages	6	20.7%	8	28.6%
≥6 minimum wages	7	24.1%	2	7.1%
**SMA**				
Type 1	0	0%	6	21.4%
Type 2	12	41.4%	10	35.7%
Type 3	16	55.2%	12	42.9%
Type 4	1	3.4%	0	0%

* Minimum wage in Brazil is equivalent to approximately 222.40 dollars.

**Table 2 ijerph-19-16935-t002:** Reports of positive and negative experiences related to the moment of communicating the diagnosis of spinal muscular atrophy from the perspectives of patients and family members during the semi-structured interview.

Thematic Axes	Positive Factors	Negative Factors
Clarification of the diagnosis	A. “I was lost, knowing the diagnosis made everything clearer” B. “I spent my youth trying to figure out the diagnosis, so finding out had no negative impact.” C. “At each consultation, we had information about news regarding SMA and we learned everything necessary” D. “Doctors guided us on which way to go and were with us in each evolution” E. “She printed several pages of websites and studies, this helped me a lot, it gave me a completely different vision of the future, of life” F. “We had a list of unanswered questions, so we talked to this doctor, he gave us a class and that’s when we were the most relieved.” G. “The doctor was enlightening and kind” H. “Informed me calmly and clearly” I. “I understood a little more about SMA, that it’s not my laziness, that I don’t have the strength and I adapted better. Today I can make plans, and think a little more about the future” J. “The doctor explained everything to me, and said I had to be patient and strong, she has a clearer, popular vocabulary that makes herself understood”.	a. “The doctor was in a hurry, he only said that she had SMA and that there was no cure, the rest I found on Google ” b. “I received a copy of the report by email, I was alone, I was helpless, desperate” c. “I did not understand the diagnosis, the information was not clear” d. “I received a correspondence with my diagnosis, who read it was a doctor from my city who had no experience and scared the whole family” e. “The doctor said: we are investigating spinal muscular atrophy and I said: OK and that’s how we got out of there and I googled the name of the disease” f. “I received the result by email. The exam had technical terms that I researched and didn’t understand, so being welcomed at the time of diagnosis is very important, maybe preparation beforehand or someone to break the news, not just a paper or a screen or just typed information” g. “The neurologist said it was a disease that had no cure, and he doesn’t know why I wanted to know” h. “The family did not understand what the term “progressive and degenerative” was, they should use a term that is easy to understand. The diagnosis scares, but when there is no understanding, the family is left completely lost” i. “There should be a preparation, but he simply spoke the diagnosis, the information passed was very cold, I felt desperate in the way” j. “The doctor was very well prepared, more scientific. She used terms I couldn’t understand. I left desperate and ran to the internet *”* k. “I had a lot of doubts. My mother stayed with me, so the questions were directed at her and not at me.” l. “It took me a while to find out what it really was, but when I was 15, I looked it up on the internet, I was terrified” m. “My diagnosis was given in my childhood, but my mother was illiterate, she didn’t understand, I only started my treatment in adolescence, when I went to a pulmonologist, who explained it to me”.
2. Communication of the prognosis	A. “They said what was going to happen, but they also said that there were many possibilities, that the cognitive part was not affected” B. “You can do anything you want. Study, go to college, work, get married, have children, not giving me false hope, but explaining to me that the future could bring me good news.” C. “Today is a challenge, but 10 years from now there will be a lot of good things happening and that comforted my heart” D. “He encouraged me to fight, to seek what was best for him.”	a. “I was told that my life expectancy was up to 15 years and that until I died, I would be in a vegetative state.” b. “Our son was going to die by the age of 10 and there was nothing to do. The doctor left and told the resident to reveal the diagnosis.” c. “I got an expiration date” d. “You will stop talking and walking, at each appointment, they only said that I would get worse” e. “I was only four years old, I don’t remember well, but it was something like: she will be a vegetable, she will not eat and will depend on appliances” f. “I thought I was going to die in my early teens and my mother kept crying” g. “He won’t talk, he won’t talk, he has a life expectancy of 2 years maximum and he also said that miracles don’t exist, so make the most of it” h. “He has a degenerative and progressive disease and a maximum of seven years of survival, at that moment he started talking and I understood nothing anymore” i. “He will have to live inside the hospital. A summary of everything was made on the first day, how my life was going to be, until the possibility of his birthday being inside the hospital”.
3. Affective memory related to the diagnosis process	A. “Today I feel welcomed and assisted” B. “They are true angels” C. “The doctor told me to trust him and explained to me about the disease. I calmed down” D. “He welcomed us, so we feel so safe that you leave the appointment and say thank you” E. “The doctor understood mothers, and put herself in their place, so she is very good” F. “She calmed us down, a very dear, welcoming, and careful professional”. G. “The doctor was very affectionate, delicate, and hugged me. She was human”.	a. “What specialist am I looking for? I asked. He replied: buy a wheelchair! I almost fainted in the living room” b. “A doctor said: don’t love him, he won’t live over eight years and I said: How can I not love him? He’s my son” c. “The doctor was very cold and cruel” d. “I should have been more careful with words, more caution, more compassion” e. “It lacked sensitivity, love for others, I think empathy sums it up” f. “Enjoy your son! This is frustrating, it’s like saying there’s nothing you can do, it was blunt and dry, not giving any kind of hope.” g. “I heard that I shouldn’t have children, as if I were a bad seed that couldn’t be spread around the world” h. “‘Are you still walking?’ We always were very apprehensive because of hearing this” i. “Come back here if you need to, if she gets worse” j. “The doctor was cold as if she had found another SMA on the way, very quickly, I didn’t feel welcomed, but I didn’t feel disrespected either. You can’t cry! she said. And my tear flowed, I couldn’t control it. He’s my son, he can be whatever he is, we love him a lot, and that’s not why we don’t cry and smile. I am human” k. “ The doctor should have welcomed me in his room, and talked a little, there was a lack of humanity, reception, someone to hug and give this information, which is not easy to give and never will be, but I was helpless” l. “I felt like I was on a reality show ”
4. Advice to doctors who reported the diagnosis.	A. “Listening to the patient is essential” B. “You have to stop, sit down, and explain” C. “You must be cautious when communicating, be reassuring and have compassion” D. “Be careful when talking, without losing hope, putting yourself in the other’s shoes, seeing not only a patient but a person who seeks help, who is trying to save a child, who is running out of time” E. “You can be realistic, give comfort, you don’t have to lie to soften the disease, but talk about what is necessary and what the possibilities are” F. “Professionals must show that they have other skills and are not afraid to pass on information, to teach, to be empathetic and human” G. “Explain the disease, how it works and what it entails. It’s better to know than to discover little by little that it gets worse without really understanding it” H. “Even if we know it’s going to hurt, we need to feel embraced by those who are going to pass the message” I. “Be straightforward about explaining the disease and its progress, what is possible and what is not” J. “Talking to the patient even if he has a companion on his side, even if he is a child or elderly person, no matter the age, but sit with the patient and patiently explain until he understands” K. “The professional should provide the information and welcome the family. It is important that it is not just the mother or the father, but the family as a whole. An initial conversation and always with this positive preparation of not dealing with the absence of a cure, but with the possibility of treatment and with the quality of life, because we focus on the good side, trying to look at the positive side” L. “I appreciate the care, the clarity, the frankness, for not having treated me in a childish way, the reception, the communication, the calm, the professionalism with which they gave me the information”.	

## Data Availability

Not applicable.

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
