# Peer review of "Communication of the Diagnosis of Spinal Muscular Atrophy in the Views of Patients and Family Members, a Qualitative Analysis"

_ijerph, 2022, doi:10.3390/ijerph192416935_

Round 1

Reviewer 1 Report

Communication of the diagnosis of spinal muscular atrophy in 1 the view of patients and family members, a qualitative analysis, which has been submitted by Isabella Araujo Mota Fernandes et al. to IJERPH, contains important opinions of patients with spinal muscular atrophy (SMA) and their family members.

Regrettably, it is difficult for the researchers outside Brazil to fully understand the manuscript because the medical care for SMA patients in Brazil is not described in the manuscript.

I think the communication style of SMA diagnosis can be changed before and after introduction of genetic testing and new drugs for SMA (ex. nusinersen etc). But the authors did not describe enough whether genetic testing is performed in Brazil, or whether new drugs have already been introduced in Brazil.

Minor issues

It is necessary to explain some abbreviated words such as HULW-UFPB and AME. AME may stand for “atrofia muscular espinhal”. As this manuscript is written in English, it is also necessary to explain the abbreviated words in English.

Author Response

Dear Reviewer, thank you for your comments. The following information was added to the methodology in order to better contextualize the studied reality: “Responses were obtained from patients and family members from public and private services in equal proportions. Such reports have described the range from the pre-genetic testing era to the present day. Currently, genetic tests are available in the private system and in some specialized public services. During this research, drugs for the treatment of SMA were acquired only through legal means, but there are three of them in the process of incorporation.”

The AME abbreviations have been changed to SMA and HULW-UFPB to Lauro Wanderley University Hospital, Federal University of Paraiba.

Reviewer 2 Report

Dear Authors, 

It was pleasure to can review your article because is saying about very important clinical subject. 

Out of concern for its quality, I make the following suggestions:
- line 18-19 - Fifty-seven volunteers from all regions of Brazil  reported positive and negative experiences about clarifying the diagnosis, communicating the prognosis, affective memory related to the event, in addition to advice to doctors - please reword this sentence so that it actually reflects the research results

- line 24-27 - please formulate the sentence in such a way that it presents the conclusions drawn from the results of your research

- line 28 - please edit the keywords correctly

- Introduction section - please structure the introduction and discuss in it the issues relevant to the implementation of the purpose of the work in accordance with its title

- please specify the aim of the work

- Please read the article thoroughly and remove typographical errors and repetitions e.g. The relevance of the topic discussed is justified is justified by the lack of technical

- Consider changing the word "volunteers" 

- In future research, please consider adapting the research tool in such a way that it is possible to conduct analyzes in terms of both quantitative and qualitative, and not only having a mainly descriptive nature

- Conslusion section - please reword the conclusions so that they are indeed conclusions resulting from your research and not a repetition of the results

- Limitations section - please consider the problem related to the research tool 

Thank you, 

Your R 

Author Response

Comments and Suggestions for Authors

- line 18-19 - Fifty-seven volunteers from all regions of Brazil reported positive and negative experiences about clarifying the diagnosis, communicating the prognosis, affective memory related to the event, in addition to advice to doctors - please reword this sentence so that it actually reflects the research results

- line 24-27 - please formulate the sentence in such a way that it presents the conclusions drawn from the results of your research

Answer line 18-19 and line 24-27:

Dear Reviewer, thank you for your comments. We have made a change to the entire summary to accommodate the suggestions. If further modifications are needed, send me the suggestions so that I can improve the text.

“Methods: This qualitative study was developed through semi-structured interviews, via teleconsultation. The analysis was developed by systematically condensing the answers and synthesizing them into four thematic axes (clarification of the diagnosis, communication of the prognosis, affective memory related to the event, and advice to physicians). Results and discussion: Twenty-nine patients with SMA and 28 family caregivers of people with this condition, from all regions of Brazil, reported that individualized, clear, honest, and welcoming communication, emphasizing positive aspects, in the presence of family members and with the possibility of continuous monitoring, were important to meet their communication needs. While lack of empathy, monitoring and guidance, and estimating life expectancy resulted in negative experiences. Conclusion: The communication needs of patients and family members described during the clarification of the diagnosis and prognosis of SMA predominantly involve empathic factors related to the attitude of the attending physician throughout the evolution of the disease. Future research evaluating other neurodegenerative diseases, as well as the development of research protocols, are important to improve communication between physicians, patients, and family members.”.

- line 28 - please edit the keywords correctly

Answer:

Keywords:

spinal muscular atrophy; physician-patient relations; communication; diagnosis

- Introduction section - please structure the introduction and discuss in it the issues relevant to the implementation of the purpose of the work in accordance with its title

- please specify the aim of the work

Answer:

Thank you for your comments, modifications have been made.

- Please read the article thoroughly and remove typographical errors and repetitions e.g. The relevance of the topic discussed is justified is justified by the lack of technical

Answer:

Thank you, a thorough review has been made of the article to check for any errors or repetitions.

- Consider changing the word "volunteers" 

Answer:

This word was modified throughout the text.

- In future research, please consider adapting the research tool in such a way that it is possible to conduct analyzes in terms of both quantitative and qualitative, and not only having a mainly descriptive nature

Thanks for the suggestion, it was added to the limitations section.

- Conclusion section - please reword the conclusions so that they are indeed conclusions resulting from your research and not a repetition of the results

Answer:

“This study has shown that individual and family communication needs during the clarification of the diagnosis and prognosis of SMA predominantly involve empathic factors throughout the evolution of the disease. Negative experiences in communicating the diagnosis of SMA, described by patients and their mothers, warn of the need to improve and adapt techniques and protocols for breaking bad news among neurologists 29,30,31, in order to strengthen the millenary relationship between physicians and the patient, we must favor the application of ethical principles in medical practice and, consequently improve short and long-term therapeutic responses, as well as adherence to ongoing care for these patients. This research should be expanded to other neurodegenerative diseases, as well as be used as a basis for developing and validating research protocols to improve communication between physicians, patients, and families”.

- Limitations section - please consider the problem related to the research tool 

Answer:

 “The absence of a validated research tool for this purpose is another limitation of this research.”

Reviewer 3 Report

Thank you to all three authors for your work on this very important and valuable topic area.  I enjoyed reading your paper and I believe, with clarifying edits, it will contribute to advancing the knowledge of doctor-patient communication for population in South America, particularly in the context of genetic/neurodegenerative diseases.  

Suggestions for review:

Abstract: 

Lines 13-14: Clarify/define “transmission” – transmission of prognosis information, or information about diagnosis alone; rephrase the paragraph to emphasize the direction of “transmission of information” and who the main “transmitter” is – e.g., the healthcare provider. Clarify how the patient-doctor binomial includes “the family” …

Lines 15-16 Objective: Clarify/define “quality of disclosure” and the source of this disclosure

Lines 16-17: Avoid repetition of “carried out”; revised English language for this paragraph

Lines 19-20: It is a little confusing what do you mean by “positive and negative experiences about clarifying the diagnosis, communicating the prognosis, affective memory related to the event, in addition to advice to doctors”. Doctor’s advice includes all the elements mentioned as “in addition to”; Clarify “positive and negative experiences”- what were the exact outcomes you are referring to- expand and include the 4 main thematic axes from Table 2. Revie English language for this paragraph. 

Lines 22-23: “communication demands” – from whom? Is “demands” the appropriate term for this? Align these outcomes with the Objective, Methods and Results stated above (include them in the definition of “quality of disclosure” ). Clarify what the “barriers” are to. 

Lines 24-27 Conclusion: Be specific what “Failures in the transmission” mean; Be specific about “improvement and adequacy of communication techniques and protocols”- such as? 

Add why this is important, apart from “the application of ethical principles in medical practice”- for example it impacts patients’ healthcare outcomes (long and short term), compliancy to treatment, follow-up on care, etc. 

Lines 31-32: Rephrase this sentence- it is confusing talking about understanding the “clinical history” – who is doing the understanding, the patient or the doctor, and adding “doing good and avoiding evil” in the same sentence– this is a vague reference to medical ethics; The term “evil” is not a great choice. 

Lines 37-39: The traditional paternalistic practice of medicine is not done with the “justification” of reducing patients’ autonomy. These are implicit consequences of this type of practice. Please revise paragraph.

Line 46: You mention long term repercussions for children- expand to include the patients (children and adults) as well. 

Lines 47-48: Expand that these are the “communication failures you mention in the previous sentence”. 

Line 52-58 Consider moving the disease definition at the beginning of your paper

Line 53: Define” quality of the dissemination”; Further, clarify “dissemination of diagnosis”- be specific who is doing the dissemination -if you are only referring to dissemination of information from the provider. I would have 2 separate paragraphs: one for the definition of “SMA” and one for the Aim(s) of the paper. Be specific – either choose “dissemination of diagnosis “for all the paper, or “dissemination of prognosis” for all your paper- avoid interchanging these terms back and forth as they have different meanings. 

Line 59: Revise repeating “justified”. This is a new idea that you are introducing: “lack of training”- it has not been mentioned up to this point in your paper (it is not mentioned at all in your Abstract), yet it appears that this is one of your main reasons for conducting this study. Rephrase as either a main reason and include it in all previous relevant parts of the paper OR rephrase the justification – focusing on improving patient’s healthcare outcomes via bridging gap in the doctor-patient communication and suggest additional training as a possible solution to this problem. 

Line 61: What do you mean by “ethical understanding of the communication needs”? Do you mean “clear understanding” or are you trying to say something different? 

It isn’t clear if by “ethical understanding” you equate that with  clear communication with the patient.  Is a physician acting unethically if they lack good communication skills? (Which the author points out that lack of training may be a cause for this issue.  Then is it the medical programs acting unethically?). 

Lines 62-63 says strengthening doctor-patient relationship.  If the mother is considered the patient for an under 18 years old child than why the use of “family members information” throughout the paper.

Lines 69-70: What associations or databases have you used- be specific; include years for this data sets

Line 96: states the mean age was 36 yet age is not provided in the table. If the data was looked at in aggregate, combined patients and SMA mothers a total column may be useful with the same headers (n and %). 

Lines 97: clarify “mothers present at the time of diagnosis” – for patients diagnosed when they were minors or for all age groups you are reporting? Also, it appears that all family were mothers, so please consider revising the idea of using “family” as a generalized term throughout the paper.

Lines 99-100: clarify “heard opinions from 3 or more doctors”- where these seen in appointments by these doctors, how did they “hear” these multiple opinions? 

Lines 100-102: Rephrase, emphasizing that 4 Thematic axes emerged from the analysis, with mixed positive and negative factors for the patient’s experience with provider’s prognosis communication.

Table 1: Not clear what “<3 minimum wage, between 3-5 minimum wages and >= 6 minimum wages “mean. Please add explanatory footnote to the table.  Include which minimum wage are you considering – Brasil’s or other. 

Education in the table would make more sense in ascending order

Line 140: You are mentioning just the children diagnosed- consider expanding to adult patients as well. This aspect should be clarified throughout the paper- you are going back and forth between just children with SMA, or a mix of both adults and children. This line/paragraph discusses what the literature says about communication with affected children but not what the authors found in this study. 

Line 145: Define AME announcement 

 Line 197: Conclusion: You are only mentioning the disclosure of SMA diagnosis, while your study includes communication of prognosis as well. Please revise and adjust throughout the paper. 

Line 205-207: Expand to the important positive changes in patients’ health outcomes as primary reasons for improved doctor-patient communication (in addition to the ethical practice of medicine). 

Author Response

Cover letter

Reviewer 3

Comments and Suggestions for Authors

Suggestions for review:

Abstract: 

Lines 13-14: Clarify/define “transmission” – transmission of prognosis information, or information about diagnosis alone; rephrase the paragraph to emphasize the direction of “transmission of information” and who the main “transmitter” is – e.g., the healthcare provider. Clarify how the patient-doctor binomial includes “the family” …

Answer:

Thank you for your comments and suggestions. It was modified in the text:

The communication needs of patients and family members described during the clarification of the diagnosis and prognosis of SMA predominantly involve empathic factors related to the attitude of the attending physician throughout the evolution of the disease.”

Lines 15-16 Objective: Clarify/define “quality of disclosure” and the source of this disclosure

Answer:

The communication needs of patients and family members described during the clarification of the diagnosis and prognosis of SMA predominantly involve empathic factors related to the attitude of the attending physician throughout the evolution of the disease.”

Lines 16-17: Avoid repetition of “carried out”; revised English language for this paragraph 

Answer: Thank you for your comments, modifications were made.

Lines 19-20: It is a little confusing what do you mean by “positive and negative experiences about clarifying the diagnosis, communicating the prognosis, affective memory related to the event, in addition to advice to doctors”. Doctor’s advice includes all the elements mentioned as “in addition to”; Clarify “positive and negative experiences”- what were the exact outcomes you are referring to- expand and include the 4 main thematic axes from Table 2.

Answer: Thank you for your comments, modifications were made for the abstract and overall text.

In the abstract

Methods: This qualitative study was developed through semi-structured interviews, via teleconsultation. The analysis was developed by systematically condensing the answers and synthesizing them into four thematic axes (clarification of the diagnosis, communication of the prognosis, affective memory related to the event, and advice to physicians). Results and discussion: Twenty-nine patients with SMA and 28 family caregivers of people with this condition, from all regions of Brazil, reported that individualized, clear, honest, and welcoming communication, emphasizing positive aspects, in the presence of family members and with the possibility of continuous monitoring, were important to meet their communication needs. While lack of empathy, monitoring and guidance, and estimating life expectancy resulted in negative experiences..

In the text

“Patients and family caregivers answered the following questions: “How did you receive the SMA diagnosis?”, “What was negative in communicating the diagnosis?”, “What was positive in communicating the diagnosis?”, “How would you like to have received the SMA diagnosis?” and “What advice would you send to health professionals who participated in your journey through the diagnosis of SMA?” and, finally, “If there is anything about the moment of receiving the SMA diagnosis that you want to tell me, I am eager to hear about it””.

Lines 22-23: “communication demands” – from whom? Is “demands” the appropriate term for this? Align these outcomes with the Objective, Methods and Results stated above (include them in the definition of “quality of disclosure”). Clarify what the “barriers” are to. 

Answer: “The communication needs of patients and family members described during the clarification of the diagnosis and prognosis of SMA predominantly involve empathic factors related to the attitude of the attending physician throughout the evolution of the disease.”

I have used ‘needs’ instead of ‘demands’. It’s a better fit.

Lines 24-27 Conclusion: Be specific what “Failures in the transmission” mean; Be specific about “improvement and adequacy of communication techniques and protocols”- such as? 

Answer:

I have made changes to the conclusion: “The communication needs of patients and family members described during the clarification of the diagnosis and prognosis of SMA predominantly involve empathic factors related to the attitude of the attending physician throughout the evolution of the disease.”

Add why this is important, apart from “the application of ethical principles in medical practice”- for example it impacts patients’ healthcare outcomes (long and short term), compliancy to treatment, follow-up on care, etc. 

These data were added in the final conclusion.

Lines 31-32: Rephrase this sentence- it is confusing talking about understanding the “clinical history” – who is doing the understanding, the patient or the doctor, and adding “doing good and avoiding evil” in the same sentence– this is a vague reference to medical ethics; The term “evil” is not a great choice.

Answer:

“The importance of communication between doctors and patients, for the benefits of health care, has been described since the time of Hippocrates1

Lines 37-39: The traditional paternalistic practice of medicine is not done with the “justification” of reducing patients’ autonomy. These are implicit consequences of this type of practice. Please revise paragraph.

Answer:

“Traditional paternalistic medicine, in which the opinion of the physician prevails, is still frequent in clinical practice 4,5, reducing the freedom of opinion and autonomy of patients 2,3”

Line 46: You mention long term repercussions for children- expand to include the patients (children and adults) as well. 

Answer:

“with long-term repercussions for patients and family caregivers with genetic disorders”

Lines 47-48: Expand that these are the “communication failures you mention in the previous sentence”. 

Line 52-58 Consider moving the disease definition at the beginning of your paper

Line 53: Define” quality of the dissemination”; Further, clarify “dissemination of diagnosis”- be specific who is doing the dissemination -if you are only referring to dissemination of information from the provider. I would have 2 separate paragraphs: one for the definition of “SMA” and one for the Aim(s) of the paper. Be specific – either choose “dissemination of diagnosis “for all the paper, or “dissemination of prognosis” for all your paper- avoid interchanging these terms back and forth as they have different meanings.

Answer lines 47-48, 52-58:

Modifications have been made.

“      

Spinal muscular atrophy (SMA) is a rare, genetic, neurodegenerative, and disabling disease, whose symptoms can appear from early childhood to early adulthood. It results in the dysfunction and death of lower motor neurons, leading to a decrease in appendicular, axial, and medullary strength, culminating in difficulty in swallowing, speaking, breathing, and maintaining cognitive integrity and sensory and sensorial functions 14.

Respecting autonomy and adequate communication of the diagnosis and prognosis of neurodegenerative and disabling diseases, such as SMA, impacts the relationship between physicians, patients, and family caregivers, the mental health of those involved, and the therapeutic follow-up 8,9,13. This helps to avoid anguish and suffering 8,9, with short and long-term repercussions for patients and family caregivers with genetic disorders 10,11.

In recent years, dissatisfaction has been observed regarding emotional support, time spent in medical consultations, and the nature and quality of information on neurodegenerative diseases 12. The need for communication standards centered on patients 5 and/or their families 8,10 are strategies that respect and protect their autonomy.

 Line 59: Revise repeating “justified”. This is a new idea that you are introducing: “lack of training”- it has not been mentioned up to this point in your paper (it is not mentioned at all in your Abstract), yet it appears that this is one of your main reasons for conducting this study. Rephrase as either a main reason and include it in all previous relevant parts of the paper OR rephrase the justification – focusing on improving patient’s healthcare outcomes via bridging gap in the doctor-patient communication and suggest additional training as a possible solution to this problem. 

Answer:

“This research aims to understand which aspects of communicating the SMA diagnosis are considered positive and negative experiences in the view of patients and their family caregivers, to improve the healthcare experience, since there is a gap in the literature related to communicating this diagnosis, causing moderate to intense stress among neurologists 12.”

Line 61: What do you mean by “ethical understanding of the communication needs”? Do you mean “clear understanding” or are you trying to say something different? 

It isn’t clear if by “ethical understanding” you equate that with clear communication with the patient.  Is a physician acting unethically if they lack good communication skills? (Which the author points out that lack of training may be a cause for this issue.  Then is it the medical programs acting unethically?). 

Answer: Thank you for your comments. For better understanding, we have changed the text to:

“This research aims to understand which aspects of communicating the SMA diagnosis are considered positive and negative experiences in the view of patients and their family caregivers, to improve the healthcare experience, since there is a gap in the literature related to communicating this diagnosis, causing moderate to intense stress among neurologists 12.”

Lines 62-63 says strengthening doctor-patient relationship.  If the mother is considered the patient for an under 18 years old child than why the use of “family members information” throughout the paper.

Answer:

“the relationship between physicians, patients, and family caregivers”

Lines 69-70: What associations or databases have you used- be specific; include years for this data sets

Answer:

“Sampling was of the simple random type; and contacts were obtained during the study period, through the Regional and National Association of Carriers of Rare Diseases, Association of Neuromuscular Diseases of Paraiba (DOENMUS), Friends of AME (AAME), and Association of Families and Friends of Carriers of Neuromuscular Diseases (DONEM). “

Line 96: states the mean age was 36 yet age is not provided in the table. If the data was looked at in aggregate, combined patients and SMA mothers a total column may be useful with the same headers (n and %). 

Answer:

“The mean age of the patients was 32.86 years (range: 19-46) and the mean age of the mothers was 39.43 years (range: 24-53).”

Age information has been added to the table

Lines 97: clarify “mothers present at the time of diagnosis” – for patients diagnosed when they were minors or for all age groups you are reporting? Also, it appears that all family were mothers, so please consider revising the idea of using “family” as a generalized term throughout the paper.

Answer:

Reports from mothers present at the time of diagnosis were accepted, regardless of the age of the patient.

Lines 99-100: clarify “heard opinions from 3 or more doctors”- where these seen in appointments by these doctors, how did they “hear” these multiple opinions? 

Answer:

“Until the definitive diagnosis, 36 (63%) respondents heard opinions from three or more doctors in public and private services”

Lines 100-102: Rephrase, emphasizing that 4 Thematic axes emerged from the analysis, with mixed positive and negative factors for the patient’s experience with provider’s prognosis communication.

Answer:

 Modifications were made

Table 1: Not clear what “<3 minimum wage, between 3-5 minimum wages and >= 6 minimum wages “mean. Please add explanatory footnote to the table.  Include which minimum wage are you considering – Brasil’s or other.

Answer:

*1 minimum wage in Brazil is equivalent to approximately 222.40 dollars

Education in the table would make more sense in ascending order

Answer:

Correction made in the table           

Line 140: You are mentioning just the children diagnosed- consider expanding to adult patients as well. This aspect should be clarified throughout the paper- you are going back and forth between just children with SMA, or a mix of both adults and children. This line/paragraph discusses what the literature says about communication with affected children but not what the authors found in this study. 

Answer:

“I have revised the text with the aim of expanding the discussion to patients who are adults or children.”

Line 145: Define AME announcement 

Answer:

SMA announcement

 Line 197: Conclusion: You are only mentioning the disclosure of SMA diagnosis, while your study includes communication of prognosis as well. Please revise and adjust throughout the paper. 

Answer:

This study has shown that individual and family communication needs during the clarification of the diagnosis and prognosis of SMA predominantly involve empathic factors throughout the evolution of the disease”

Line 205-207: Expand to the important positive changes in patients’ health outcomes as primary reasons for improved doctor-patient communication (in addition to the ethical practice of medicine). 

Answer:

“in order to strengthen the millenary relationship between physicians and the patient, we must favor the application of ethical principles in medical practice and, consequently improve short and long-term therapeutic responses, as well as adherence to ongoing care for these patients”

.”

Round 2

Reviewer 1 Report

The authors thoroughly revised the first version. The second version is now much better than the first one.

As I commented in the first review report, Communication of the diagnosis of spinal muscular atrophy in the view of patients and family members, a qualitative analysis, which has been submitted by Isabella Araujo Mota Fernandes et al. to IJERPH, contains important opinions of patients with spinal muscular atrophy (SMA) and their family members. I felt more like that after I read the revised version of the manuscript.

Minor concerns

(1) Hippocrates in the Introduction section (Line 34)

I think it is not necessary to mention Hippocrates. I think that “without Hippocrates” may be better than “with Hippocrates” for the following three reasons. 

First, the idea of autonomy in the Hippocratic Oath has been controversial. Some say that paternalism originated in the Hippocratic Oath, and others say that the idea of autonomy came from the Hippocratic Oath. There are diametrically opposed opinions. Thus, citation of Hippocrates in the Introduction section may confuse your readers.

Second, the quotation style of Hippocratic Oath is not proper in the Reference section of the current version. Two sources are shown under one reference number. They should be shown separately. If you would like to stick to Hippocrates”, it would be necessary to revise this part.

Third, without Hippocrates, your story remains solid. You can start the Introduction section with the second paragraph. (But this is just my suggestion.)

(2) AME in the Methodology section (Line 76)

The authors changed AME and HULW-UFPB to SMA and Lauro Wanderley University Hospital, Federal University of Paraiba, respectively. This revision enables the doctors outside Brazil to understand this manuscript more easily. I think that “Friends of AME (AAME)” in the 76th line can be revised into AAME Friends of SMA (AAME)”.

(3) Drugs in the Methodology section (Lines 81-82)

The authors wrote “three drugs” in the following sentence. “During this research, drugs to treat SMA were acquired only through legal means, but there are three of them in the process of incorporation.”

I think it is better to show the names of the three drugs in the sentence like this; “During this research, new drugs to treat SMA could be acquired only through legal means, but there are three of them (nusinersen, onasemnogene abeparvovec-xioi, and risdiplam) in the process of incorporation.”

(4) Family members with minimum wage in Table 1

I think that “family members” should be “family numbers”. (But I am not sure. Am I wrong?)

Author Response

Dear Reviewer, thank you for your comments.

Minor concerns

(1) Hippocrates in the Introduction section (Line 34)

I think it is not necessary to mention Hippocrates. I think that “without Hippocrates” may be better than “with Hippocrates” for the following three reasons. 

First, the idea of autonomy in the Hippocratic Oath has been controversial. Some say that paternalism originated in the Hippocratic Oath, and others say that the idea of autonomy came from the Hippocratic Oath. There are diametrically opposed opinions. Thus, citation of Hippocrates in the Introduction section may confuse your readers.

Second, the quotation style of Hippocratic Oath is not proper in the Reference section of the current version. Two sources are shown under one reference number. They should be shown separately. If you would like to stick to “Hippocrates”, it would be necessary to revise this part.

Third, without Hippocrates, your story remains solid. You can start the Introduction section with the second paragraph. (But this is just my suggestion.)

Thank you for your comments, the sugested modifications have been made. I deleted Hippocrates from the first paragraph.

(2) AME in the Methodology section (Line 76)

The authors changed AME and HULW-UFPB to SMA and Lauro Wanderley University Hospital, Federal University of Paraiba, respectively. This revision enables the doctors outside Brazil to understand this manuscript more easily. I think that “Friends of AME (AAME)” in the 76th line can be revised into “AAME Friends of SMA (AAME)”.

Thank you for your comments, the sugested modifications have been made.

(3) Drugs in the Methodology section (Lines 81-82)

The authors wrote “three drugs” in the following sentence. “During this research, drugs to treat SMA were acquired only through legal means, but there are three of them in the process of incorporation.”

I think it is better to show the names of the three drugs in the sentence like this; “During this research, new drugs to treat SMA could be acquired only through legal means, but there are three of them (nusinersen, onasemnogene abeparvovec-xioi, and risdiplam) in the process of incorporation.”

Thank you for your comments, the sugested modifications have been made.

(4) Family members with minimum wage in Table 1

I think that “family members” should be “family numbers”. (But I am not sure. Am I wrong?)

I changed it to "Family Income"

Reviewer 2 Report

Dear Authors, 

Thank you for considering the suggestions - I hope they were useful.

Kind regards

Author Response

The suggestions were very important.
Thank you very much for your attention to my article.